# HIV status alters immune cell infiltration and activation profile in women with breast cancer

Marcus Bauer[1,2] ✉, Pablo Santos [2], Andreas Wilfer[1,3], Eunice van den Berg[4], Annelle Zietsman[5], Martina Vetter [6], Sandy Kaufhold[6], Claudia Wickenhauser[1], Isabel dos-Santos-Silva[7], Wenlong Carl Chen [2,8,9,10], Herbert Cubasch[11], Nivashini Murugan [11], Valerie McCormack[12], Maureen Joffe[2,8,13,14], Barbara Seliger [15,16,17] ✉ & Eva Kantelhardt[2,6]

The breast cancer (BC)-related mortality is higher and the immunity is altered in women living with HIV (WLWH) compared to HIV-negative women. Therefore, tumor samples of 296 black BC patients from South Africa and Namibia with known age, HIV status, tumor stage, hormone receptor and HER2 status and overall survival (OS) are analyzed for components of the tumor microenvironment (TME). WLWH ($n = 117$), either with suppressed viral activity (HR = 1.25) or with immune suppression (HR = 2.04), have a shorter OS. HIV status is associated with increased numbers of CD8$^+$ T cells in the TME compared to HIV-negative patients; no correlation is found with CD4$^+$ T cell numbers in the blood. Moreover, an increased expression of CD276/B7-H3 and a more pronounced IFN-γ signaling in the tumors are found in WLWH, independent of age, stage, and BC subtypes. In conclusion, altered T cell composition and CD276 expression in WLWH may contribute to inferior survival and can be used for targeted treatment.

Female breast cancer (BC) is the most common type of cancer worldwide and an increasing public health concern in sub-Saharan Africa (SSA)[1–3]. Due to a weaker health infrastructure with predominantly diagnosis at an advanced stage and limited treatment options, the survival of BC patients in SSA is particularly low when compared to patients in Europe or North America[4–6]. In addition to this, differences in the BC biology including a distinct regional variation in the distribution of BC subtypes, genetic aberrations and the tumor microenvironment (TME) have been reported with unfavorable prognostic factors in Western Africa[7–9]. In Southern and Eastern Africa, an additional factor influencing BC management and survival is the relatively high population prevalence of human immunodeficiency virus (HIV)[10] that influences the tumor biology and patients' tolerability and response to therapy and survival outcomes.

There are large age- and sex-specific as well as a regional variations in HIV prevalence across SSA[11] with Namibia, South Africa,

Lesotho and Botswana exhibiting the highest prevalence[12]. The development of highly active anti-retroviral therapy (HAART) was a major breakthrough and has dramatically improved patients' survival[13]. In recent years, the availability of HAART was accompanied by a substantially reduced AIDS related mortality[14]. Consequently, women living with HIV (WLWH) live to ages when BC incidence rates in the population are rising. Thus, WLWH and BC represent a substantial proportion of the BC patient profile in these Southern African countries[15,16].

Recently, comparison of WLWH with BC and HIV negative BC patients demonstrated a younger age at diagnosis and poorer overall survival of WLWH (HR = 1.49). The inferior survival in WLWH was independent of age, stage and the hormone receptor status[15–17], but a higher prevalence of TNBC was reported recently[18]. Of note, both WLWH with <50 viral load copies/mL and WLWH with ≥50 viral load copies/mL had a poorer survival than HIV negative BC patients[17]. In

addition, a small study reported that 17 people living with HIV who developed a malignancy in the course of the HIV disease showed an increased T cell dysfunction and exhaustion prior to cancer diagnosis[19]. This was accompanied by an increased expression of immune checkpoint (ICP) molecules, such as the programmed death receptor 1 (PD-1) as a major regulatory factor upon HIV infection[20]. These data suggest a complex HIV associated immune deregulation even upon HAART. However, the influence of the HIV infection on the composition of the TME and the expression of immune-relevant molecules on tumors has not yet been analyzed in detail.

Therefore, the objectives of this study are to compare the immune cell composition and functionality as well as the expression of immune modulatory molecules between WLWH and HIV negative BC patients, overall and stratified by age, stage, and hormone receptor status, which might give further insights into the lower survival rate of WLWH with BC. A better understanding regarding the function of the cellular and non-cellular components of the TME contributing to tumorigenesis and disease progression of BC in WLWH will have an impact on the intervention and treatment of those patients.

## Results

### Association of the HIV status with the clinical stage, IHC groups, and BC patients' OS

In order to analyze the influence of HIV status on BC biology we compared clinical and biological variables in relation to HIV status (Fig. 1) (Table 1). Of the 117 WLWH, 10% were Namibian, whereas of the 179 HIV negative BC patients, 37% were Namibian. This strong HIV-country association of HIV status implies that crude differences between HIV positive and HIV negative patients may be due to differences between the countries and not due to HIV infection, unless country-specific models are presented or unless there is an adjustment for the country.

From those 117 WLWH with BC, 47 patients showed low CD4$^+$ T cell counts (<500 cell/mm$^3$) and detectable viral loads in the blood at the time point of diagnosis. Those individuals were categorized as WLWH immune suppressed (WLWH - IS), while WLWH with higher counts of CD4$^+$ T cells and no detectable viral loads were classified as WLWH virally suppressed (WLWH - VS). Principal component analysis (PCA) showed an overlap in features of BC specimens from WLWH - IS and WLWH - VS, while the samples of HIV negative BC patients were partially separated (Fig. 2A). Comparison of clinical variables revealed

a younger mean age in WLWH with BC at time of diagnosis (Fig. 2B, Table 1) as expected. Moreover, no differences regarding proportions of TNBC and HER2 + BC were found between WLWH and HIV negative BC patients (see Table 1 and Fig. 2C, D). Next, we analyzed OS in 291 patients. Univariate analysis (Fig. 2E) revealed a higher risk in patients with higher numbers of children (HR = 1.08, 95% CI:1.00–1.18), younger age (HR = 1.12, 95% CI: 1.01–1.21), advanced clinical stage (HR = 3.24, 95% CI: 2.11–4.98), HER2 + BC and TNBC (HR = 2.01, 95% CI: 1.11–3.65 and HR = 1.77, 95% CI: 1.13–2.79, respectively). WLWH - IS showed worse outcome (HR = 2.04, 95% CI:1.17–3.56) compared to WLWH - VS. Patients, who received endocrine treatment or radiation therapy showed better survival.

### Frequency of TILs is associated with the HIV status

The frequency of TILs in the TME had no significant prognostic value (Fig. 2E) in the studied BC patient cohort (representative H&E staining images in Fig. 2F). A mean higher TIL frequency was found in HER2+ and TNBC compared to the Luminal A-like BC (p = 0.002). Moreover, a higher TIL density was found in WLWH - VS compared to HIV negative patients (Fig. 2G), while we found no statistically significant difference between WLWH – IS and WLWH – VS or WLWH – IS and HIV negative BC patients. In order to understand the prognostic value of tumor intrinsic factors, HIV status and abundancy of TILs, a multivariate Cox regression analysis including country of origin, number of children, clinical stage, IHC groups, HIV status, treatment, and the density of TILs was performed. As shown in Fig. 2H, the hazard ratio for WLWH - VS was reduced from 1.25 in the univariate analysis to 0.95 in the multivariate analysis, while it decreased in WLWH - IS BC patients from 2.04 to 1.34. Adjustment for IHC groups was the factor attenuating the HIV effect on patients' outcome. Higher frequencies of TILs showed no statistically significant prognostic value after adjustment.

### Aberrant T cell infiltration in the BC TME of WLWH

Despite only differences in the TIL density were found in tumors of WLWH - VS versus HIV negative BC patients, the composition of the immune cell subpopulations as well as the expression of different immune relevant molecules, like ICPs and HLA class I antigens, was determined. First, the bulk RNA expression of the whole tumor including its TME (n = 296) was analyzed for the expression profile of 41 different immune genes (Supplementary Table 1) employing an unsupervised clustering model as shown in Fig. 3. The clusters based

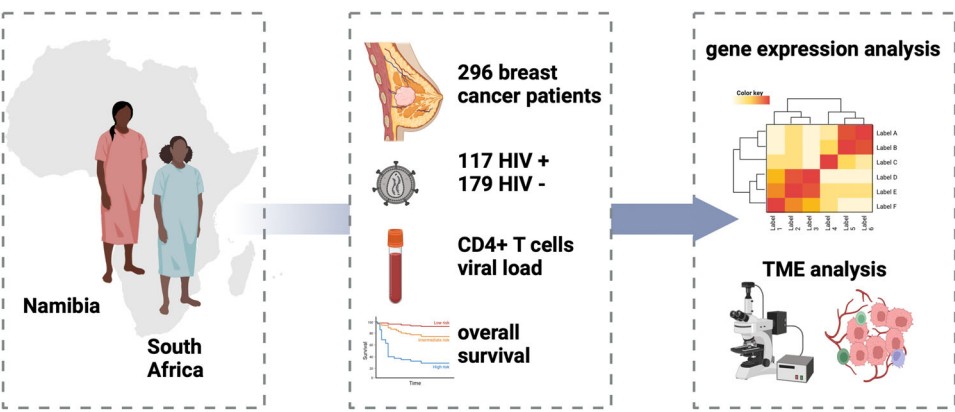

**Fig. 1 | Graphical abstract.** Tumor samples of 296 black women with breast cancer (BC) from South Africa and Namibia and known age at diagnosis, HIV status, clinical tumor stage, BC hormone receptor and HER2 status as well as overall survival (OS) were analyzed for components of the BC tumor microenvironment (TME). The frequency of tumor infiltrating lymphocytes (TILs), immune cell subpopulations and the expression profile of immune modulatory molecules were analyzed by histomorphology, RNA expression analysis and multiplex immunohistochemistry (MSI). Women living with HIV (WLWH) (n = 117) had a shorter OS, both WLWH with suppressed viral activity (HR = 1.25) and with immune suppression (HR = 2.04). HIV status was associated with increased numbers of CD8$^+$ T cells in the TME compared to HIV negative BC patients, which did not correlate with CD4$^+$ T cell numbers in the blood. Moreover, an increased protein expression of immune checkpoint molecule CD276/B7-H3 and a more pronounced IFN-γ signaling in the tumors were found in WLWH, independent of age, stage, and BC subtypes. The graphical abstract was created in BioRender (Bauer, M. (2025) https://BioRender.com/aaf1liv).

**Table 1 | Clinical and immunological parameters in breast cancer specimens and patients (n = 296)**

| Parameters | | WLWH - VS | WLWH - IS | HIV negative |
|---|---|---|---|---|
| Number of patients | | 70 | 47 | 179 |
| Country of origin | South Africa (n) | 58 | 47 | 113 |
| | Namibia (n) | 12 | 0 | 66 |
| Age (mean) | | 27-81 (48) | 30-81 (49) | 29-98 (57) |
| Stage (simplified) | Early | 24 (34.3%) | 5 (10.6%) | 62 (34.6%) |
| | Advanced | 46 (65.7%) | 42 (88.4%) | 117 (65.4%) |
| Stage (UICC) | I | 3 (4.2%) | 0 (0%) | 7 (3.9%) |
| | II | 45 (64.3%) | 24 (51.1%) | 88 (49.2%) |
| | III | 18 (25.7%) | 19 (40.4%) | 63 (35.2%) |
| | IV | 4 (5.8%) | 4 (8.5%) | 21 (11.7%) |
| Grading | G1 | 2 (2.9%) | 2 (4.3%) | 15 (8.5%) |
| | G2 | 35 (50.0%) | 25 (53.2%) | 90 (51.1%) |
| | G3 | 33 (47.1%) | 20 (42.6%) | 71 (40.4%) |
| Ki67 index | <25% | 24 (34.3%) | 10 (21.3%) | 76 (43.2%) |
| | ≥25% | 46 (65.7%) | 37 (88.7%) | 100 (56.8%) |
| IHC groups | Luminal A-like | 18 (25.7%) | 4 (8.5%) | 62 (34.7%) |
| | Luminal B-like | 36 (51.4%) | 29 (61.7%) | 72 (40.2%) |
| | HER2+ | 7 (10.0%) | 6 (12.8%) | 16 (8.9%) |
| | TNBC | 9 (12.9%) | 8 (17.0%) | 29 (16.2%) |
| TILs | <10% | 27 (38.6%) | 22 (46.8%) | 85 (48.3%) |
| | 10-39% | 32 (45.7%) | 22 (46.8%) | 70 (39.8%) |
| | ≥40% | 11 (15.7%) | 3 (6.4%) | 21 (11.9%) |
| T cell Infiltration | CD3$^+$(mean) | 8.2% | 7.5% | 3.6% |
| | CD3$^+$CD8$^-$(mean) | 2.3% | 1.2% | 1.1% |
| | CD3$^+$CD8$^+$(mean) | 4.7% | 5.6% | 1.9% |
| | CD3$^+$FoxP3$^+$(mean) | 1.2% | 0.7% | 0.6% |
| T cell proximity (µm) | CD3$^+$T cell→CD8$^+$ T cell (mean) | 24 µm | 27 µm | 56 µm |
| | CD3+T cell→Treg (mean) | 72 µm | 88 µm | 105 µm |
| | CD8+T cell→Treg (mean) | 59 µm | 86 µm | 32 µm |
| | CD3+T cell→ panCK$^+$ cancer cell (mean) | 31 µm | 30 µm | 29 µm |

*BC* breast cancer, *HER2 +* human epidermal growth factor receptor 2 positive, *IHC* immunohistochemistry, *TILs* tumor infiltrating lymphocytes, *TNBC* triple negative breast cancer, *WLWH - IS* women living with *HIV – immune suppressed*, *WLWH - VS* women living with *HIV – viral suppressed*.

on the expression of the 41 different genes did neither associate with the HIV status, nor with other parameters, such as the clinical stage, tumor grading, tumor cell proliferation Ki67 index, and the IHC groups.

However, univariate explanatory analysis of all immune genes indicated candidate genes, namely CD4, CD8, and CD276, to be associated with the HIV status. In order to investigate the impact of the HIV status on the expression of these genes in the BC TME, we next analyzed the interrelationship of HIV status on the expression of CD4, CD8, and CD276 (Fig. 4A–F) demonstrating a lower CD4 expression in WLWH - VS, but a trend of higher CD8 expression in WLWH - IS. In WLWH - IS, a trend of higher CD276 expression was also detected. Multivariate analysis including socio-economic factors and treatment are shown for all three markers in the Supplementary Figs. 1–3. Next, Pearson correlation of different TME factors revealed a correlation of CD276 RNA expression with CD4, CD56 and CD68, while Tregs (FOXP3), B cells (CD20) and granzyme B (grB) showed an inverse correlation (Fig. 4G). In addition, the same analysis was performed separated by the HIV status (Supplementary Figs. 4–5). Moreover, an independent prognostic value for CD4 and CD276 RNA expression was found in a multivariate Cox regression (Fig. 4H). Higher CD4 expression was associated with improved patients' outcomes (HR = 0.607, 95% CI: 0.41–0.89), while higher CD276 expression was linked with worse outcomes (HR = 1.596, 95% CI: 1.01–2.55).

Association of an increased CD276 expression, interferon gamma (IFN-γ) signaling and T cell exhaustion is associated with an aberrant spatial T cell distribution in the BC TME of WLWH

Since a prognostic impact for CD276 was found and the CD4 and CD8 gene expression were associated with HIV status, 48 BC specimens, containing 24 cases of WLWH (8 WLWH–VS, 16 WLWH–IS) and 24 cases of HIV negative patients were subjected to MSI using two

8-plex panels including CD3, CD8, FoxP3, IDO-1, PD-1, LAG-3, CTLA4, and CD69, as well as panCK, CD3, pStat1, ISG15, HLA-I heavy chain (HC), HLA-G, TAP1, and CXCR4. First, the different T cell populations (CD3$^+$CD8$^-$ T cells, CD3$^+$CD8$^+$ T cells and CD3$^+$FoxP3$^+$ regulatory T cells (Tregs)) were analyzed as representatively shown in Fig. 5A. While the percentage of CD3$^+$CD8$^-$ T cells (Fig. 5B) was comparable in WLWH and HIV negative patients, CD3$^+$CD8$^+$ T cell numbers were higher in WLWH overall, but particularly WLWH - IS (p = 0.033, Fig. 5C). The frequency of CD3$^+$FoxP3$^+$ Treg was higher in WLWH - VS (Fig. 5D). Pearson correlation analysis revealed no association of the frequency of TILs, CD3$^+$CD8$^-$ / CD3$^+$CD8$^+$ T cells or CD3$^+$FoxP3$^+$ Tregs and the number of CD4$^+$ T cells in the peripheral blood in WLWH as well as in T subpopulations like CD3$^+$CD8$^-$ and CD3$^+$CD8$^+$ T cells (Fig. 5E–H). The spatial proximity of CD3$^+$CD8$^-$ and CD3$^+$CD8$^+$ T cell subsets was increased in the TME of WLWH. Moreover, WLWH - VS showed a closer spatial proximity of CD3$^+$CD8$^-$ and CD3$^+$CD8$^+$ T cells to CD3$^+$FoxP3$^+$ Tregs (Fig. 5I). Next, the expression of CD276 was analyzed by IHC as representatively shown in Fig. 5J and revealed a higher CD276 protein expression in WLWH - IS (Fig. 5K). The interrelationship of TIL density, CD276 expression and the numbers of different T cell subpopulations is shown in Fig. 5L. CD276 expression was not associated with CD4$^+$ T cell numbers in the peripheral blood and TIL density, but correlated with higher numbers of CD3$^+$CD8$^+$ T cells in the BC TME (Fig. 5M–O). Interestingly, a spatial heterogeneity of CD276 expression was found within the tumors that was associated with regional differences of TIL densities (Fig. 6). In addition, the expression of different immune response relevant molecules was analyzed as representatively shown in Fig. 7A (for all antigens refer to Supplementary Fig. 6). The expression of both, CTLA4 and LAG3 was increased in different T cell subsets, while the PD-1 expression was only slightly higher in WLWH (Fig. 7B–D) and no difference was found in the CD69 surface expression

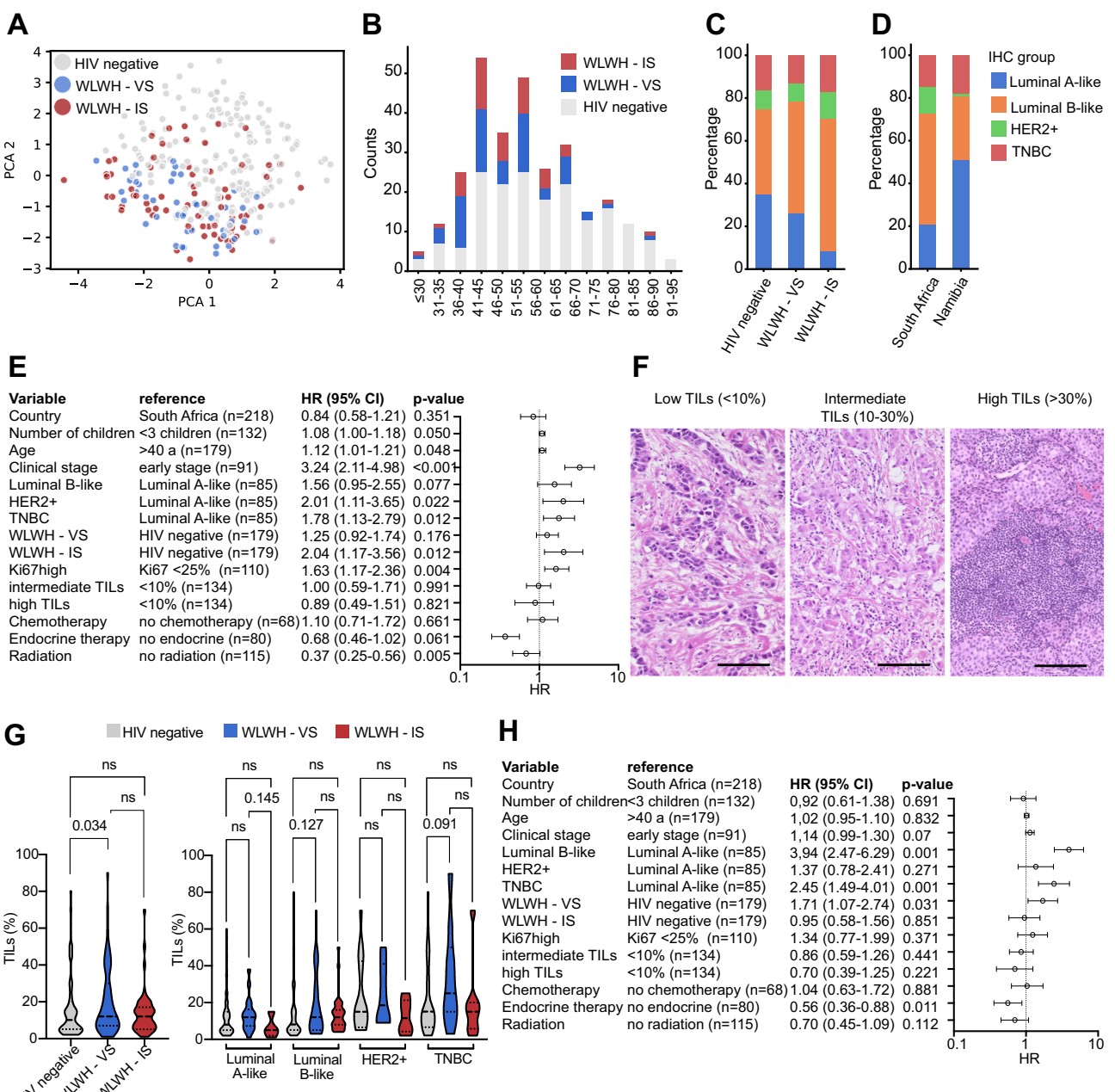

**Fig. 2 | Breast cancer biology and overall survival in sub-Saharan Africa. A** PCA of BC samples and clinical features from HIV negative and WLWH BC patients. **B** Association of age with HIV status. **C, D** Distribution of IHC groups with regard to HIV status and country of origin. **E** Univariate Cox regression of 296 BC patients regarding their risk of death. The number of patients in each reference group, the hazard ratio and the 95% confidence interval, as well as the p-value are given. **F** Representative H&E staining of FFPE BC samples with low, intermediate and high (from left to right) infiltration, scale bars depict 50 μm. TIL scoring was performed by two pathologists independently. **G** Box plots show the distribution of TIL numbers in different IHC groups and regarding HIV status. The mean is given as a dashed line and the standard deviation is given as a dot line. Categories were compared using the Mann–Whitney U test and p-values are given as numbers if $p < 0.05$. **H** Multivariate Cox regression of 296 BC patients regarding their risk of death. The number of patients in each reference group, the hazard ratio and the 95% confidence interval, as well as the p-value are given. Abbreviations: BC breast cancer, HER2 + human epidermal growth factor receptor 2 positive, PCA principal component analysis, TILs tumor infiltrating lymphocytes, SA South Africa, WLWH - IS women living with HIV – immune suppressed, WLWH - VS women living with HIV – viral suppressed.

(Supplementary Fig. 7A). However, a higher expression of IFN-γ signaling pathway component ISG15 and a trend to higher pSTAT1 were found in WLWH - VS (Fig. 7E, F). ISG15 expression was not associated with CD4$^+$ T cell numbers in the blood, but was linked with higher CD276 expression (Supplementary Fig. 7B, C). The expression of HLA-I HC and HLA-G did not significantly differ between the respective groups (Fig. 7G, H). Higher expression of HLA-I HC was associated with a trend of higher CD3$^+$CD8$^+$ T cells in the TME and lower CD4$^+$ T cell numbers in the peripheral blood (Supplementary Fig. 7D, E). The protein expression of TAP1, IDO-1, and CXCR4 showed no differences (Fig. 7I, Supplementary Fig. 7F, G). To validate the higher expression of IFN-γ signaling pathway components, the publicly available RNA expression data set GSE149156[21] was employed. Analysis of bulk RNA-seq data from 6 BC specimens of WLWH versus samples from 3 HIV negative BC patients demonstrated 230 significantly downregulated and 285 significantly upregulated genes (Fig. 7J). By focussing on

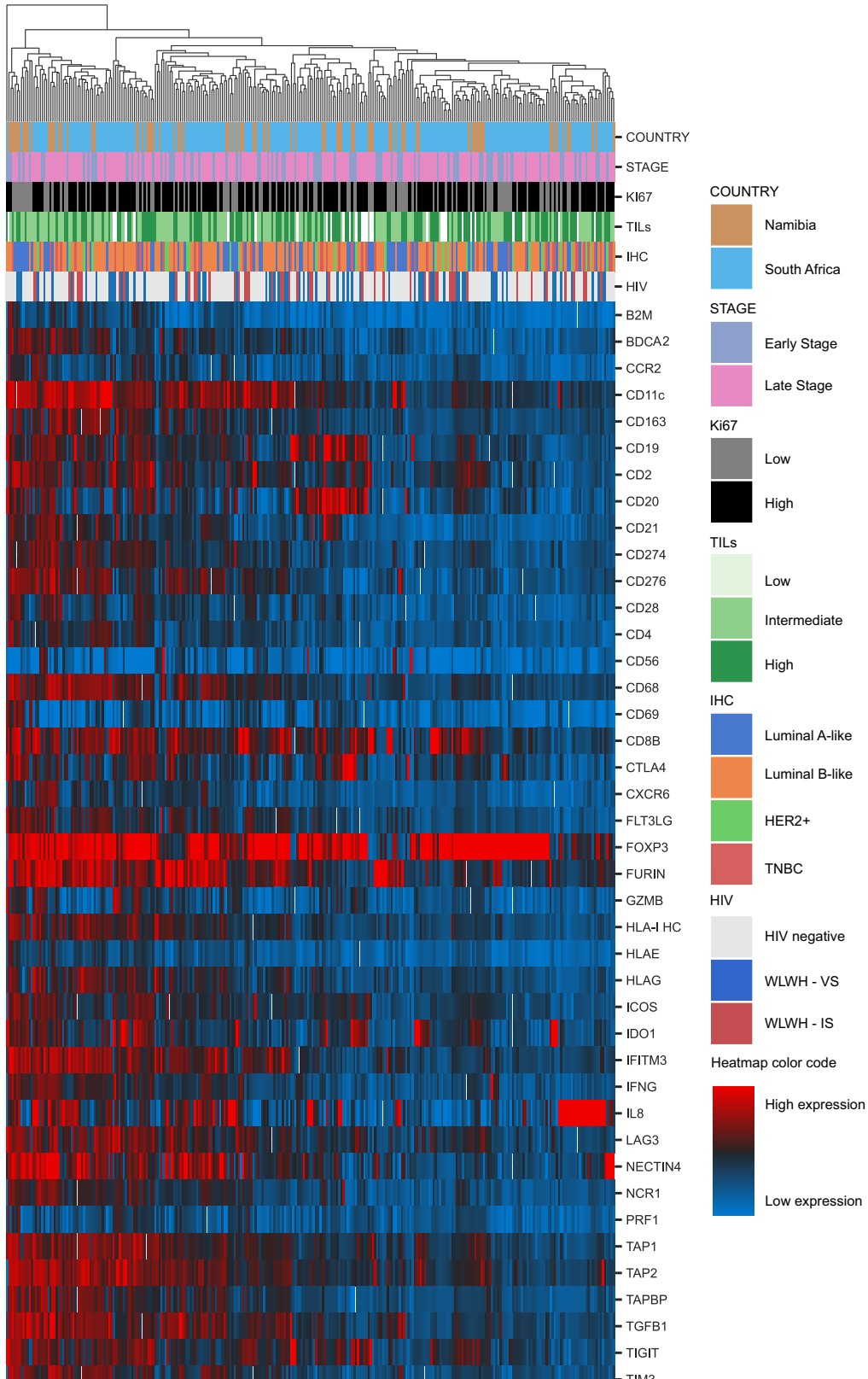

**Fig. 3 | Immune gene expression in breast cancer specimens.** Unsupervised clustering of the RNA expression of 41 genes used for the TME classification of 296 BC samples with known HIV status of patients' is given as a heatmap. Red tiles denote increased expression, while blue tiles correspond to decreased expression. The six horizontal bars above the heatmap indicate the classification of samples according to country, stage, Ki67, TILs, IHC groups and HIV status. Color codes for each bar and relative expression are given at the right side of the figure. Abbreviations: BC breast cancer, HER2 + human epidermal growth factor receptor 2 positive, IHC immunohistochemistry, TILs tumor infiltrating lymphocytes, TNBC triple negative breast cancer, WLWH - IS women living with HIV – immune suppressed, WLWH - VS women living with HIV – viral suppressed.

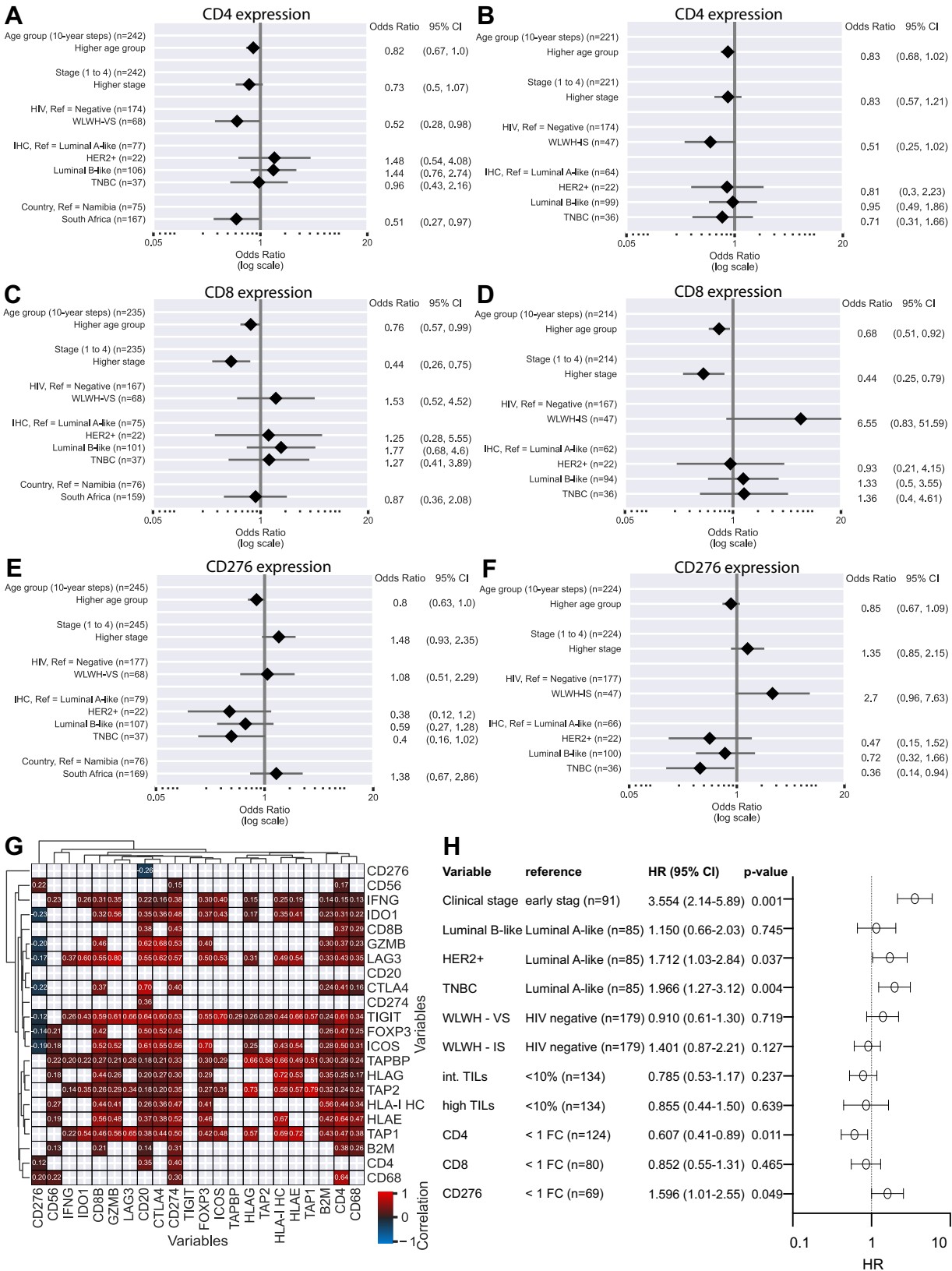

**Fig. 4 | Distinct RNA expression profile of immune-relevant genes in breast cancer samples of WLWH and HIV negative patients. A–F** Association between HIV status and CD4, CD8 and CD276 RNA expression in WLWH – VS and WLWH–IS versus HIV negative BC patients, in a multivariate analysis. **G** Correlation map of selected immune genes of all patients, irrespective of HIV status. The correlation values are shown in different colors with red tiles denote a positive correlation, while blue tiles correspond to a negative correlation. **H** Multivariate Cox's regression with stage, IHC groups, HIV status, CD4, CD8, and CD276 expression of 296 patients regarding their risk of death. The number of patients in each reference group, the hazard ratio and the 95% confidence interval, as well as the p-value are given. Abbreviations: FC fold change, WLWH – IS women living with HIV – immune suppressed, WLWH – VS women living with HIV – viral suppressed.

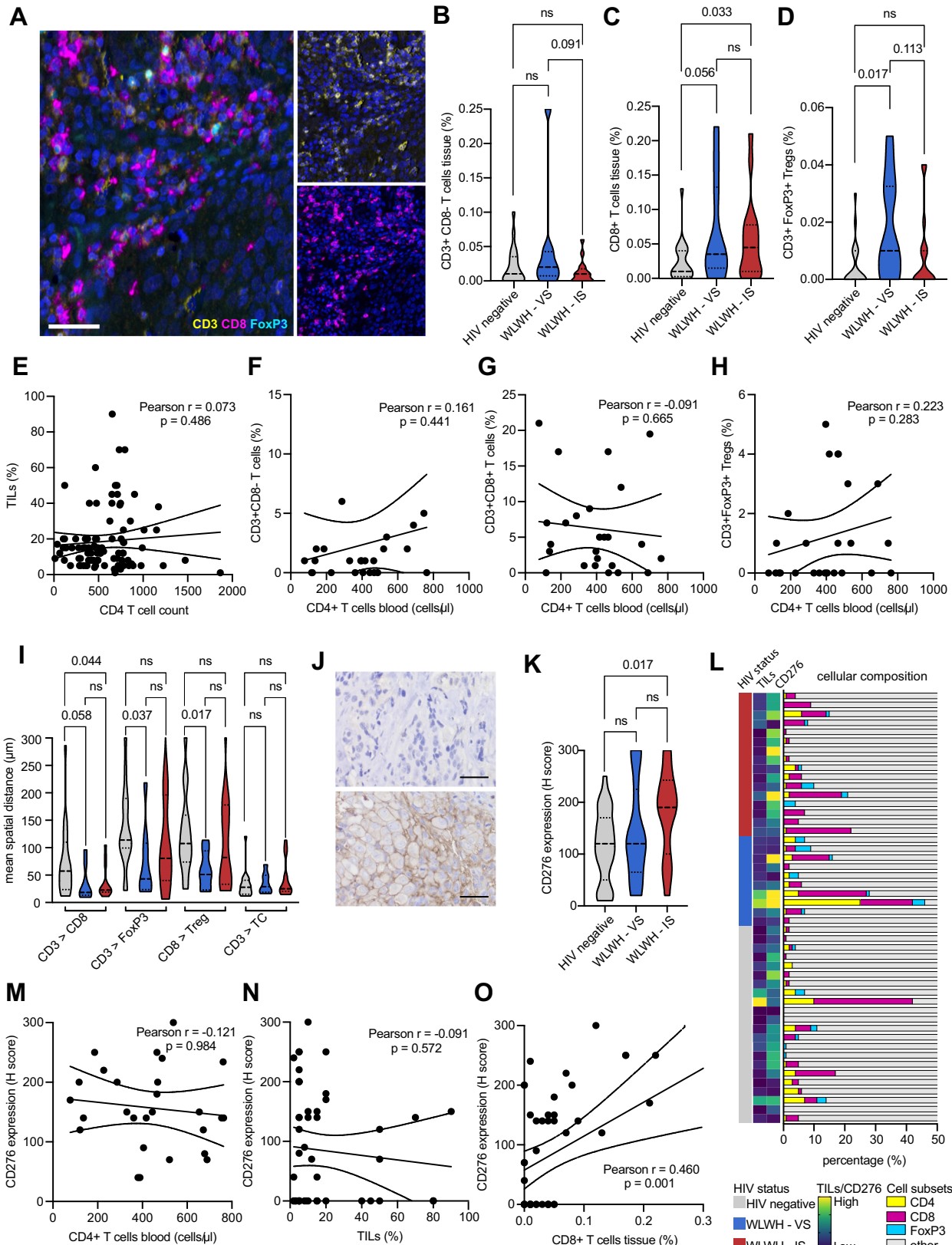

immune-related genes (Fig. 7K, genes summarized in Supplementary Table 2), an upregulation of CD14, IFNAR2 and CD55 could be demonstrated, while OAS1 and CD8B showed no significant upregulation. To get insights in the expression of different biologically relevant gene sets a GSEA was performed employing the

MSigDB_Hallmark_2020 gene set. This revealed 24 significantly upregulated gene sets in BC specimens of WLWH including the inflammatory response, TNF-α signaling via NF-kB, TGF-β signaling, IFN-α and IFN-γ response, while only the myc target V1 gene set was downregulated (Fig. 7L and Supplementary Fig. 8).

**Fig. 5 | Distinct immune cell composition in the tumor microenvironment of HIV positive and HIV negative breast cancer patients. A** Multiplex IHC using an Ab panel directed against CD3 (yellow), CD8 (magenta), FoxP3 (turquoise) and counterstained with DAPI (blue). The scale bar depicts 50 μm. For the analysis from each sample three areas were selected and cell counts and marker expression was extracted. **B–D** The amount of T cell subsets in the BC TME of HIV negative patients and WLWH. The mean is given as a dashed line and the standard deviation is given as a dot line. Categories were compared using the Mann–Whitney U test and p-values are given as numbers if *p* < 0.05. **E–H** Scatter plot, overlaid with the corresponding linear regression model and associated *p* value (two-sided) is shown for the correlation of different immune cell subsets in the TME and the numbers of CD4 + T cells in the peripheral blood. **I** The spatial distribution (mean distance of CD3 + T cells and CD8 + T cells; FoxP3+ Tregs and CD3 + T cells, FoxP3+ Tregs and CD8 + T cells as well as CD3 + T cells and panCK+ cancer cells) was analyzed and is shown with box plots. The mean is given as a dashed line and the standard deviation

is given as a dot line. **J** Representative picture of a CD276 immunohistochemistry with a strong stained and a negative case. **K** CD276 was analyzed by two pathologists independently using an H score and the expression in WLWH and HIV negative patients is shown in box plots. The mean is given as a dashed line and the standard deviation is given as a dot line. **L** Bar graph shows the percentage of CD3 + CD8- (blue), CD3 + CD8+ (red), CD3+FoxP3+ (yellow), and all other cells (gray, e.g., stroma, tumor cells) separated by HIV status and is association with TIL numbers in the stroma evaluated on the H&E stained, as well as the CD276/B7-H3 expression analyzed by IHC. **M–O** Association of CD276 expression and CD4 + T cell numbers in the peripheral blood, TILs in the TME, and CD3 + CD8+ is shown with a scatter plot, overlaid with the corresponding linear regression model and associated p-values (two-sided) are given. Abbreviations: TILs tumor infiltrating lymphocyte, WLWH - IS women living with HIV – immune suppressed, WLWH - VS women living with HIV–viral suppressed.

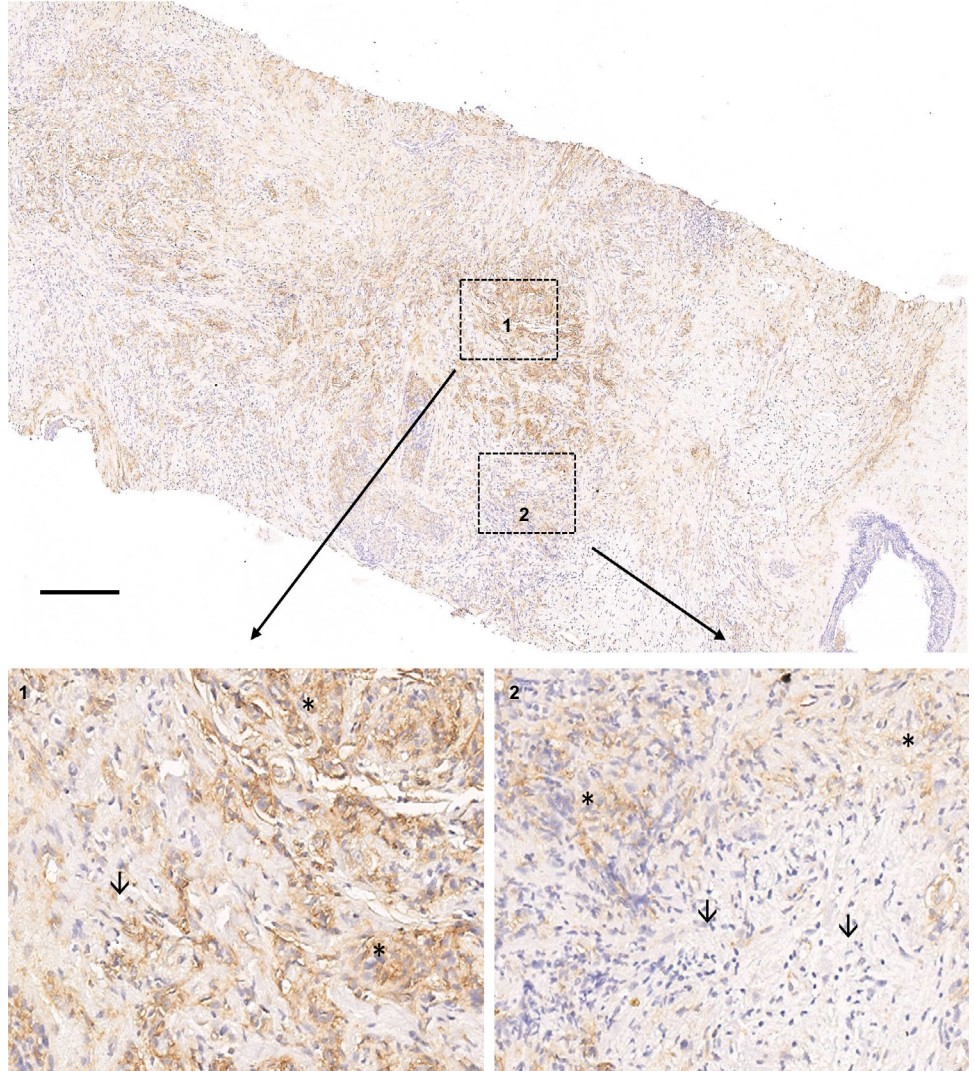

**Fig. 6 | CD276 protein expression in the BC TME.** Example of a CD276 staining of a BC sample of a HIV positive patient showing a spatial heterogeneity with a high expression in region 1 in comparison to region 2 with a lower CD276 expression. Tumor cells are marked with asterisks and TILs are highlighted with arrows. The scale bar depicts 500 μm.

## Discussion

Despite the incidence of AIDS related malignancies like Kaposi sarcoma or non-Hodgkin lymphoma decreased in the era of HAART people living with HIV still have an increased cancer risk[19,22]. In addition, those patients also show a reduced overall survival when compared with HIV negative cancer patients[15]. Thus, a better

understanding of the underlying causes of this difference is urgently needed in order to develop strategies for reducing mortality among this patient group. During the last years, not only tumor intrinsic factors, but also the TME, the patients' immunity, environmental factors and (viral) infections have been shown to influence the initiation and progression of malignant diseases[23–26]. Therefore, this study addressed

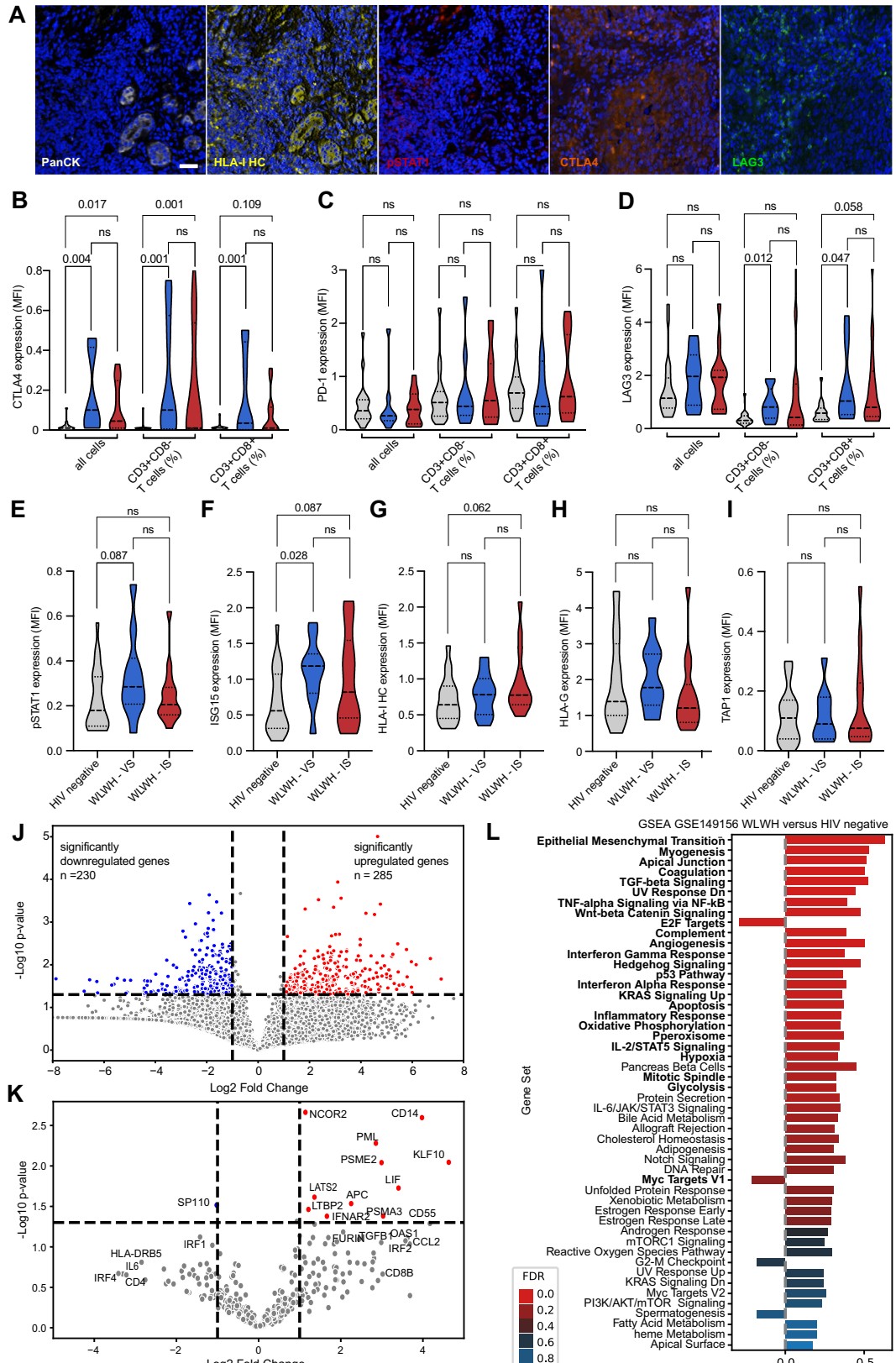

the possible association between the BC biology, immune cell composition and expression of immune relevant markers, like ICP molecules, HLA antigens, APM components and IFN-γ signaling pathway components in the TME and its interrelationship with chronic HIV infection.

For this purpose, samples from two different study cohorts of BC patients from South Africa and Namibia with known clinical course were used. As already reported, we found a younger age at diagnosis in WLWH with BC, while no differences among the proportions of HR and HER2 expression status could be detected[17]. Histopathological analysis

**Fig. 7 | ICP expression and IFN-γ signaling in BC. A** Representative pictures of panCK+ cancer cells (gray), HLA-I HC (yellow), pStat1 (red), CTLA4 (orange), and LAG3 (green). The scale bar depicts 50 μm. **B–D** The expression of ICP molecules CTLA4, PD-1 and LAG3 were tested on different T cell subpopulations. The mean is given as a dashed line and the standard deviation is given as a dot line. Categories were compared using the Mann–Whitney U test and p-values are given as numbers if $p < 0.05$. **E–I** The expression of pStat1, ISG15, HLA-I HC, HLA-G and TAP1. The mean is given as a dashed line and the standard deviation is given as a dot line. Categories were compared using the Mann–Whitney U test and p-values are given as numbers if $p < 0.05$. **J** Volcano plot depiction of differentially expressed genes in WLWH versus HIV negative BC (GSE149156) for all genes and **K** immune related genes. Significantly upregulated genes are marked in red, significantly downregulated genes in blue. **L** Bar plot depiction of GSEA (MSigDB_Hallmark_2020) of WLWH versus HIV negative. The NES is given as the value of the bars and the FDR is depicted in different colors. Gene sets with a significant up-/down-regulation are highlighted with bold letters. Abbreviations: CXCR4 CXC motif chemokine receptor 4, CTLA4 cytotoxic T-lymphocyte-associated protein 4, GSEA gene set enrichment analysis, HLA human leucocyte antigen, ISG15 Interferon-stimulated gene 15, LAG3 lymphocyte-activation gene 3, panCK pan-cytokeratin, PD-1 programmed cell death protein 1, TAP transporter of antigen processing, WLWH women living with HIV.

of the TME by determining the TIL densities in the BC stroma revealed that this well-established biomarker[27] showed no predictive relevance in this patient cohort and that WLWH – VS showed the highest mean TIL abundancy. Therefore, an exploratory analysis of the gene expression of 41 immune genes was carried out and revealed an upregulation of CD8 and CD276, but a downregulation of CD4 in the TME of WLWH. Of note, higher CD8 expression in BC samples from WLWH has been previously associated with improved outcome in BC[28], but high numbers of exhausted T cells predicted shorter survival in some BC subtypes[29]. Since the HIV infection preceded BC in WLWH, the influence of HIV status on the expression of the immune genes was analyzed demonstrating differences in the CD4, CD8, and CD276 RNA expression between BC samples from HIV negative individuals versus WLWH with viral suppression and/or immune suppression. Considering there exists not always a coordinated mRNA and protein expression, the expression of immune relevant molecules was also determined by MSI and/or conventional IHC[30,31]. Moreover, the selected samples were balanced for the HIV status and IHC group and downstream analyses were only performed with samples from South Africa in order to avoid a country specific bias.

MSI confirmed the mRNA data demonstrating an aberrant T cell infiltration of the TME with an increased abundance of CD8+ T cells. Despite these data were not related to an increased IFNG mRNA expression in tumor tissues, an activation of the canonical IFN-γ signaling pathway was demonstrated with an increased Stat1 phosphorylation and ISG15 expression[32]. This discrepancy might be due to the fact that only local IFNG expression levels were analyzed. Under physiological conditions, IFN-γ is secreted by T cells and NK cells[33,34] and stimulated by interleukins[35] or pathogens, like infectious agents[36]. In the context of HIV, a strong systemic increase of IFN-γ was shown in the acute infection[37], while in the chronic disease contrary results were published ranging from decreased to sustained high serum levels of IFN-γ[38–40]. The activated IFN-γ signaling in BC tissues of WLWH was validated using publicly available RNA expression data sets[21]. Comparing RNA expression of BC specimens from WLWH and HIV negative BC patients, GSEA revealed an activation of several immune pathways including inflammatory response, TNF-α signaling via NF-kB, TGF-β signaling, IFN-α and IFN-γ response. Recently, a dual role of IFN-γ signaling with both synergistic and antagonistic effects has been described[41]. There is growing evidence that chronic IFN-γ signaling inhibits the maintenance, clonal diversity and proliferation of T cells to restrict anti-tumor immunity, while the inhibition of IFN-γ signaling leads to phenotypically less exhausted TILs, but impaired ICP inhibition[42–44]. Additionally, HLA-I antigens are regulated by IFN-γ signaling as well[45,46] and a slightly higher HLA-I expression in BC of WLWH was found in this patient cohort. Furthermore, the expression of several ICP molecules like PD-1[47], LAG3[48,49], CTLA4[50] and CD276[51,52] is increased by IFN-γ signaling. In BC of WLWH an increased expression of these markers known to be associated with T cell exhaustion can already be found years prior to the manifestation of malignant diseases[19]. In our study, a higher expression of the ICP molecule CD276 was shown in the tumor tissue, particularly in WLWH - IS, which was associated with increased expression of T cell exhaustion markers CTLA4 and LAG3 in different T cell subpopulations. It is well known

that CD276 inhibits the efficacy of CD8+ cytotoxic T cells and drives tumor immune evasion that correlated with a decreased T cell infiltration within the TME[53,54]. An inverse correlation of increased frequencies of CD38+CD39+CD4+ T cells and CD276 expression was found, which correlated with a better patients' prognosis in CD276 deficient tumors[53]. Interestingly, the more pronounced T cell exhaustion and CD276 expression in WLWH resulted in aberrant spatial T cell distribution showing a regional heterogeneity within the tumor. This was associated with a closer proximity between different T cell subsets independent from the CD4+ T cell numbers in the peripheral blood. It has been shown that the spatial proximity of different immune cell subsets is associated with the initiation and the progression of malignant tumors[55–57]. Moreover, multivariate cox regression analysis showed an independent prognostic value of both CD4 and CD276 expression. Due to its frequent overexpression, CD276 has already been identified as a valuable target for the treatment of malignant solid and hematological tumors[11,58]. Currently, a number of clinical trials using inhibitors including anti-CD276 antibody drug conjugates, or monoclonal antibodies alone or in combination with other checkpoint inhibitors and CAR-T cell therapies are ongoing[59–62].

Taken together, our data show that WLWH with BC in SSA undergoing HAART showed (i) an activated IFN-γ signaling in the TME, (ii) an upregulation of the ICP molecule CD276 and (iii) a more pronounced T cell exhaustion in particular in WLWH - IS and (iv) an aberrant T cell composition and distribution in WLWH that was independent from peripheral blood CD4+ T cell numbers. Since the global prevalence of WLWH receiving HAART is increasing over time, non-AIDS-related cancers, such as BC, have become a significant cause of death among those patients[63]. Therefore, the analysis of the TME and its influence on patients' outcome is not only interesting in areas with an outstanding high HIV prevalence like SSA, but also worldwide. Thus, an increased understanding of the functional mechanisms of T cell exhaustion in the context of cancer in general, but in particular also in cancer patients living with HIV, is an urgent need that should be addressed in future studies. This should include a more comprehensive analysis of both the TME and the systemic immune status of the patients' to enhance our understanding of the influence of chronic HIV in non-AIDS-related cancers. Furthermore, premature aging caused by the HIV infection and drug toxicity by both HAART and anti-cancer therapy including HAART-related liver damage are contributing factors that should be focused in future studies[64–66].

## Methods
### Ethical statement
This research complies with the ethical regulations. All patients provided written informed consent. Local ethical institutions (Ethical Committee of the Martin Luther University Halle-Wittenberg, Ministry of Health and Social Services of Namibia, and the Institutional Review Board of University of the Witwatersrand, Gauteng) approved the scientific use of the FFPE tissue samples (Table 2).

### Patients' characteristics and study design
We embedded this study within two BC cohorts in Southern Africa, i.e., the ABC-DO study and SABCHO[67,68], to avail of tumor specimens,

**Table 2 | Overview of the local ethical approvals**

| Institution | Country | Date of approval | Identification number |
|---|---|---|---|
| Martin Luther University, Halle-Wittenberg | Germany | 06/06/2014 | 2014-57 |
| School of Anatomical Pathology, National Health Laboratory Services | South Africa | 27/07/2014 | M140754 |
| IARC | France | — | IEC13-19, IEC15-18 |
| Ministry of Health and Social Services of Namibia | Namibia | 2017 | 17/3/3 |
| University of the Witwatersrand, Gauteng, | South Africa | — | M150345 |

clinical, epidemiological, treatment and highly complete active follow-up data. These two African BC cohorts were diagnosed during 2015–2021 and each include 2000 to 3000 BC patients newly diagnosed across multiple hospitals and for ABC-DO study, countries. For the purpose of the present study, we identified a subset of black women from two countries with a high HIV prevalence i.e., South Africa and Namibia, whose tumor specimens had been retrieved and were of good quality. We included as many BC specimens from WLWH as possible, and for comparative purposes, not all but at least as many HIV negative women. HIV status had already been ascertained at the time of diagnosis, based on self-reports in Namibia and, in South Africa, from medical records based on the enzyme-linked immunosorbent assay through the National Health Laboratory Services. In total, we analyzed formalin-fixed and paraffin-embedded (FFPE) tissue blocks of 296 women (291 with survival data), comprising 218 women from South Africa (105 HIV positive and 113 HIV negative) and 78 from Namibia (12 HIV positive and 66 HIV negative). Of this selected set, 117 tumors were from WLWH (39.5%) and 179 from HIV negative BC patients (60.5%). Clinical data available for all women include stage a diagnosis, such as TNM stage according to the Union for International Cancer Control (UICC). A simplified stage was used for tumors >5.0 cm diameter and/or with known metastasis refer to advanced stage. The CD4$^+$ T cell numbers and the viral load in peripheral blood were known from 103 patients at the time point of BC diagnosis.

### Histopathology and immunohistochemistry

Histopathological diagnoses of BC were made according to the WHO classification of Breast Tumors 5th edition[69], including histopathological grading according Elston and Ellis[70]. All samples were re-analyzed by two pathologists independently using conventional IHC using antibodies (Ab) directed against the ER, PR, HER2 and Ki67 (Supplementary Table 3). IHC staining was performed on a Bond III automated immunostainer (Leica Biosystems Nussloch GmbH, Wetzlar, Germany) using the Bond Polymer Refine Detection Kit (DS9800-CN). BC subtypes were classified according to the surrogate IHC group classification using hormone receptor expression of ER and PR, the HER2 status as well as Ki67 proliferation index including the following IHC groups: Luminal A-like, Luminal B-like, HER2 positive (HER2 + ), triple negative BC (TNBC)[71]. ER and PR negativity was defined as receptor expression of <1% of tumor cells according to the ASCO guidelines[72]. HER2 expression with DAKO score 2 (equivocal) underwent confirmatory HER2 in-situ hybridization (ISH) according to the ASCO guidelines[73]. The number of tumor infiltrating lymphocytes (TIL) was analyzed on hematoxylin and eosin (H&E) slides as described elsewhere[27] and scored by two pathologists independently.

### ICP expression and immune cell infiltration analysis

For ICP expression and TIL subpopulation analysis 48 samples from South Africa were used (equally balanced for HIV status and hormone receptor status). These samples were stained with an antibody directed against CD276/B7-H3 (details in Supplementary Table 2) according to the manufacturer's instructions. The qualitative and quantitative expression was determined employing the H score[74]. Analysis was performed by two pathologists independently.

Moreover, the frequency and localization of the different immune cell subpopulations and cancer cells as well as the expression of several immune markers was evaluated by multispectral imaging (MSI). The multispectral staining procedure was performed using standard protocols[75]. The following two antibody panels were employed: 1st panel: CD3, CD8, FoxP3, CD69, programmed cell death protein 1 (PD-1), lymphocyte-activation gene 3 (LAG-3), cytotoxic T-lymphocyte-associated protein 4 (CTLA-4), and indolamine-2,3-dioxygenase (IDO)-1; 2nd panel: CD3, Pan-cytokeratin (panCK), phosphorylated signal transducer and activator of transcription 1 (pStat1), Interferon-stimulated gene 15 (ISG15), human leukocyte antigen (HLA) class I heavy chain (HC), HLA-G, and CXC motif chemokine receptor 4 (CXCR4) (Supplementary Table 3). For imaging the PhenoImager HT (Akoya Biosciences, Marlborough, USA) was employed and three areas (each 3744 × 2808 pixel, 0.5 µm/pixel) were selected for cell segmentation and subsequent analysis. Cell segmentation and phenotyping were performed using the inform software (Version 2.6, Akoya biosciences, USA). The frequency and spatial distribution of cell populations were analyzed using an R script (https://github.com/akoyabio).

### RNA isolation and gene expression analysis

Prior to RNA isolation, FFPE specimens were microdissected. For RNA isolation, two to four 10-µm thick tissue slides were used. Deparaffinization of FFPE tissues was performed 2 times with xylene for 5 min, followed by incubation in 96% and 70% ethanol for 2 min each. Proteinase K digestion was performed for up to 2 h at room temperature followed by 15 min at 80 °C. RNA was isolated with miRNeasy Mini FFPE Kit (Qiagen, Venlo, The Netherlands) according to the manufacturer's instructions. RNA expression analysis was performed using the NanoString Assay according to the hybridization protocol for the nCounter XT CodeSet Gene Expression Assay (NanoString nCounter, Seattle, WA, USA). Raw expression data was analyzed with the nCounter Expression Data Analysis Guide (MAN-C0011-04 from 2017). Data import, quality control and normalization of expression levels were conducted with the nSolver software version 4 (NanoString Technologies) and the expression levels were evaluated with the NanoStringNorm R package (https://github.com/sgrote/NanoStringNormalizeR/). Background subtraction from raw transcript counts was achieved with negative controls. After reference normalization (dividing the geometric mean of six references genes: ACTB, G6PD, RPLP0, TBP, TFRC and UBB), expression data was log2-transformed and considered in all downstream analyses. The NCBI accession numbers of RNAs analyzed are provided in Supplementary Table 1.

### Analysis of the publicly available data set GSE149156

Normalized gene expression data of GSE149156[21] were downloaded of from Gene Expression Omnibus (https://www.ncbi.nlm.nih.gov/geo/). Differential gene expression analysis of BC specimens of WLWH (n = 6) and HIV negative patients (n = 3) was performed using PyDESeq2 package in Python[76]. For gene set enrichment analysis (GSEA), prerank function of GSEApy (PMID: 36426870) and the MSigDB_Hallmark_2020 gene set (https://www.gsea-msigdb.org/gsea/msigdb/collections.jsp) was used. Differentially expressed genes were not filtered for statistical significance before ranking. The normalized enrichment score (NES)

and the false discovery rate (FDR) were calculated for all gene sets. Volcano plots were created with python using Pandas, NumPy, SciPy and Metplotlib packages. Means and standard deviations were calculated for each gene and independent t-tests were performed. Log fold changes and −log10(p-values) were computed and the significant differentially expressed genes ($p < 0.05$, logFC > 2) were highlighted. The immune related genes for the immune focused volcano plot are summarized in Supplementary Table 2.

## Statistics

The Mann–Whitney U test was employed to compare clinical and immunological data between different HIV status and IHC groups. A multivariate logistic regression (generalized linear model, binomial family) was used to estimate odds ratios (OR) and confidence intervals (CI) concerning the effects of age (continuous variable, by year), stage (binary variable, advanced vs. early), IHC groups (categorical variable) and gene expression of selected genes ($n = 41$, Supplementary Table 3) with HIV status. Association of a positive HIV status and increasing variable values (age, or gene expression), or as compared to the reference category (stage and IHC groups with e.g., 'Luminal A-like' in the case of IHC groups) is given with OR. Regressions were performed using the Statsmodels library for Python (www.statsmodels.org). Linear correlations between continuous variables (gene expression) were estimated as Pearson's correlations. Survival analyses were performed for 291 patients (maximum follow-up time of 72 months) with differences calculated with Cox proportional hazard models (adjusted for confounding factors) regarding OS as endpoint. Except when stated otherwise, all statistical analyses were performed using IBM SPSS Statistics 25 or GraphPad Prism v9.

## Reporting summary

Further information on research design is available in the Nature Portfolio Reporting Summary linked to this article.

## Data availability

All data are included in the Supplementary Information or available from the authors, as are unique reagents used in this Article. The raw numbers for charts and graphs are available in the Source Data file whenever possible. Further data generated are available upon request to the corresponding author or ABC-DO/SABCHO study PIs. The normalized RNA expression data generated in this study have been provided in the source data file. Source data are provided with this paper.

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

## Acknowledgements

We want to thank all patients who provided tumor samples and the pathology staff. This work was supported by grants from the Else-Kröner-Foundation (EJK, 2018_HA31SP), Susan G. Komen-Foundation (EJK, GTDR16378013, V.M., IIR13264158), German Cancer Aid (B.S., Integrate-TN, 70113450), the German Research Council (BS, Se581/33-1), German Academic Exchange Service (EJK, ID57216764), Ministry for Economic Cooperation and Development and the Else-Kröner-Fresenius Foundation (EJK, ID81256434), Hoffmann-La Roche Ltd (EJK, 27.5.2014) and the National Cancer Institute (V.M., R01CA244559). Disclaimer: Where authors are identified as personnel of the International Agency for Research on Cancer / World Health Organization, the authors alone are responsible for the views expressed in this article and they do not necessarily represent the decisions, policy or views of these organisations.

## Author contributions

All authors agreed on the final version of the manuscript. The samples and clinical data were collected by E.B., A.Z., H.C., N.M., W.C.C., V.M., M.J. E.J.K., C.W., M.J., M.C. and B.S. mentored the team. Analyses were performed by M.B., P.S., A.W., S.K., M.V. and I.S.S. The original draft was written by M.B., P.S., M.J., V.M., E.J.K. and B.S.

## Funding

## Competing interests

The authors declare no competing interests.

## Additional information

¹Institute of Pathology, University Hospital Halle, Martin Luther University Halle-Wittenberg, Halle (Saale), Germany. ²Global and Planetary Health Working Group, Institute of Medical Epidemiology, Biometrics and Informatics, Martin Luther University Halle-Wittenberg, Halle (Saale), Germany. ³Krukenberg Cancer Center, University Hospital Halle, Halle (Saale), Germany. ⁴Department of Anatomical Pathology, University of the Witwatersrand, National Health Laboratory Service, Johannesburg, South Africa. ⁵AB May Cancer Centre, Windhoek Central Hospital, Windhoek, Namibia. ⁶Department of Gynecology, University Hospital Halle, Martin Luther University Halle-Wittenberg, Halle (Saale), Germany. ⁷Department of Non-Communicable Disease Epidemiology, London School of Hygiene and Tropical Medicine (LSHTM), London, UK. ⁸Strengthening Oncology Services Research Unit, Faculty of Health Sciences, University of the Witwatersrand, Johannesburg, South Africa. ⁹Sydney Brenner Institute for Molecular Bioscience, Faculty of Health Sciences, University of the Witwatersrand, Johannesburg, South Africa. ¹⁰National Cancer Registry, National Health Laboratory Service, Johannesburg, South Africa. ¹¹Department of Surgery, School of Clinical Medicine, Faculty of Health Sciences, University of the Witwatersrand, Johannesburg, South Africa. ¹²International Agency for Research on Cancer (IARC/WHO), Environment and Lifestyle Epidemiology Branch, Lyon, France. ¹³Noncommunicable Diseases Research Division, Wits Health Consortium (PTY) Ltd, University Witwatersrand, Johannesburg, South Africa. ¹⁴Strengthening Oncology Services Research Unit, Faculty of Health Sciences, University of the Witwatersrand, Johannesburg, South Africa. ¹⁵Medical Faculty, Martin Luther University Halle-Wittenberg, Halle (Saale), Germany. ¹⁶Fraunhofer Institute for Cell Therapy and Immunology, Leipzig, Germany. ¹⁷Institute of Translational Immunology, Medical School Theodor Fontane, Brandenburg an der Havel, Germany. ✉e-mail: marcus.bauer@uk-halle.de; barbara.seliger@uk-halle.de

