## [Transparent Peer Review file · Nature Communications]

HIV status alters immune cell infiltration and activation profile in women with breast cancer

Corresponding Author: Dr Marcus Bauer

Version 0:

Reviewer comments:

Reviewer #1

(Remarks to the Author)

The study investigates the TME of BC in women living with HIV (WLWH) in Southern Africa. It compares immune cell composition and immune modulatory molecule expression between HIV-positive and HIV-negative BC patients. The findings suggest that HIV-positive patients exhibit distinct TME characteristics, including higher CD8+ T cell infiltration, increased CD276/B7-H3 expression, and more pronounced IFN- γ signaling, all of which are independent of age, stage, and hormone receptor status. These alterations contribute to poorer OS in HIV-positive BC patients.

The study addresses a critical public health issue by focusing on the intersection of HIV and breast cancer, particularly in a region with high HIV prevalence.

The study utilizes a robust dataset from the ABC-DO and SABCHO cohorts, which includes detailed clinical, epidemiological, and follow-up data.

The use of histomorphology, RNA expression analysis, and multiplex immunohistochemistry (MSI) provides a comprehensive assessment of the TME.

The study provides novel insights into the immune landscape of BC in HIV-positive patients, highlighting potential biomarkers and therapeutic targets such as CD276/B7-H3.

While the sample size (296 patients) is adequate, the findings may not be generalizable beyond the specific populations studied (South Africa and Namibia).

The study mentions survival analysis but does not provide detailed statistical models or hazard ratios for various subgroups. The study controls for age, stage, and hormone receptor status but might benefit from a more detailed analysis of other potential confounders such as socio-economic status and treatment regimens.

While the study correlates immune profiles with survival, it lacks functional studies to elucidate the mechanistic pathways underlying the observed differences in the TME.

The study concludes that the altered T cell composition and CD276/B7-H3 expression in HIV-positive patients contribute to inferior survival and could be used for targeted treatment. However, it does not consider other potential factors for reduced OS such as the limited life expectancy due to HIV itself or suboptimal oncological treatment given the concomitant HIV.

These factors should be mentioned and discussed by the authors.

Recommendations:

1. Include more detailed survival analysis with multivariate models to strengthen the association between immune profiles and outcomes.
2. Discuss the findings in the context of other regions with high HIV prevalence to enhance the generalizability of the results.
3. Consider including or suggesting future studies that explore the functional mechanisms behind the altered TME in HIV-positive BC patients.
4. Discuss other potential factors contributing to reduced OS in HIV-positive BC patients, such as life expectancy limitations due to HIV and potential suboptimal cancer treatments due to concurrent HIV infection.

The manuscript presents valuable findings on the interplay between HIV infection and breast cancer, offering insights that could pave the way for targeted therapies and improved management of BC in HIV-positive patients, but it requires some additional work and clarifications to meet the high standards of Nature Communications.

These improvements will strengthen the manuscript and enhance its overall contribution to the field.

Reviewer #2

(Remarks to the Author)

Marcus Bauer et al. aimed to assess whether the HIV status influences the tumor microenvironment (TME) in breast cancer and mediates their poorer prognosis. They found HIV positive BC patients had a shorter OS, which was associated with increased numbers of CD8+ T cells, increased protein expression of CD276/B7-H3 and a more pronounced IFN- γ signaling. They provide valuable information about immune ecology of BC in HIV-positive patients. But there are also some major concern about this research.

1. In univariate COX regression analysis, "HIV positive" does not significantly predict overall survival (OS), yet "HIV positive patients with decreased numbers of CD4+ T cells" do show significant predictive power (Fig.1C). So why not utilize the latter in multivariate COX regression analysis for prediction? (Fig.1F) It is recommended that all subsequent analyses should be stratified based on this parameter.

2. Based upon the previous question, the authors state: "HIV-positive BC patients had a shorter OS (HR=1.25), which was associated with increased numbers of CD8+ T cells compared to HIV-negative BC patients." However, the abundant infiltration of CD8+ T cells was always reported to be positively associated with better survival. How to explain this controversial issue?

3. Moreover, there is no correlation between intra-tumoral CD8+ T cells and peripheral blood CD4+ T cell counts. Yet, the authors suggest that peripheral blood CD4+ T cell counts and increased numbers of CD8+ T cells are both stronger predictors of shorter OS. Is there any contradiction here? Or are there other critical factors that have not been taken into account?

4. In Fig. 3H, high expression of CD4 indicates a favorable prognosis in breast cancer, yet neither HIV status nor CD8 expression emerges as an independent predictive factor, contradicting the authors' conclusions.

5. The tumor sections require a re-evaluation for quantification. In Fig. 4I, there is no correlation between the proportion of T cell subsets and the percentage of TILs. In some cases, the total proportion of T cell subsets in a patient reaches up to 40%, while the TILs are less than 10%.

6. It is intriguing to note that HIV-positive patients exhibit a higher infiltration of CD8+ T cells within breast cancer. This phenomenon warrants further investigation. Are these CD8+ T cells antigen-specific, and do they target tumor antigens or HIV antigens?

7. HIV positive BC patients had a more pronounced IFN- γ signaling. Does the IFN- γ signaling pathway correlate with the prognosis of HIV-positive breast cancer patients?

8. Single cell sequencing and spatial profiling will provide a better understanding in the function of the cellular and non-cellular components of the TME contributing to tumorigenesis of BC in WLWH.

9. The author only analyzed the sequencing data. It's suggested to investigate the function of TME cells, especially tumor-infiltrating CD4+ and CD8+ T cells, as well as the role of CD276, in breast cancer patients with HIV. The mechanisms underlying the increased infiltration of CD8+ T cells needs to be further explored.

Reviewer #3

(Remarks to the Author)

The study by Bauer and colleagues investigated the impact of HIV infection on the tumor microenvironment (TME) of breast cancer (BC) patients in Southern Africa. The main findings suggest that women living with HIV BC patients exhibit altered immune characteristics within the TME, including higher CD8+ cell counts, increased IFN-gamma and increased CD276 expression, which may contribute to their poorer prognosis compared to women without HIV.

The study does not present original questions or findings that advance the field. Critical information regarding patient immune status is not clearly provided, as patients living with HIV and those who are indeed immunodeficient are analyzed together as a single group, potentially leading to imprecise conclusions. Additionally, changes in CD4 and CD8 expression levels in HIV patients are already well-established, and data should include blood CD4 and CD8 levels. Thus, the differences in TME observed in the study may arise due to the fact that some HIV patients are immunodeficient, exhibiting low CD4 levels and high CD8 levels. Furthermore, the connection between immune exhaustion in cancer and HIV has been previously documented, perhaps except for data on CD276 expression. Therefore, while the study reinforces known associations, it does not significantly enhance our understanding of the interplay between HIV and breast cancer.

In the general comparisons between women living with HIV versus those who are not, the best method to see potential differences was to do a case-control study, matching as much as possible (and for most variables) the two groups on a 1:3 or 1:2 basis. I don't see why the authors chose not to use this kind of study design.

In general, the manuscript has important limitations. Several "significant" associations stated by the authors did not really reached statistical significance, but that didn't discourage the authors to state them in the manuscript. Just as examples, please see hazard ratios described in Figure 1F and stated in lines 292-5 of the manuscript, PD-1 expression in Fig 5B and pStat1 expression in Fig 5C-E (listed in lines 357-61 of the ms). Differences in age of WLWH and those without infection listed in Table 1 and Fig.1A are also non-significant, yet stated as such in the text. I also recommend that authors state exact p-values in their Figures, rather than "***", so we can assess the degree of significance in the associations pointed out. This is the case of Figures 3A, B, E, G, 4C, F, H, %B, D, as well Supplementary Fig. S1.

In terms of format, the manuscript text requires extensive revision, since several errors and typos are present across the manuscript. Also of particular importance, the authors should refrain from using "HIV-infected women" or "HIV-positive women" terms, which are not currently acceptable, and rather use "women living with HIV".

Reviewer #4

(Remarks to the Author)

Version 1:

Reviewer comments:

Reviewer #1

(Remarks to the Author)
concerns have been addressed in the revisions

Reviewer #2

(Remarks to the Author)
Marcus Bauer et al. responded to all the questions and I have no further comments.

Reviewer #3

(Remarks to the Author)
The authors have made one significant change to the manuscript, which was redefining the comparison groups to distinguish fully immune recovered women from those that already started with high CD4 T-cell numbers. That, surely, had to be followed by full reanalysis of the data. That complied with issues raised by me and by Reviewer #2. On the other hand, they did not perform any additional experiments as suggested by Reviewer #1.

In general terms, I would say the manuscript improved to some extent, but some inconsistencies to what has been reported previously in the literature are still present and putative explanations are still lacking.

In cases where reviewers are anonymous, credit should be given to 'Anonymous Referee' and the source. The images or other third party material in this Peer Review File are included in the article's Creative Commons license, unless indicated otherwise in a credit line to the material. If material is not included in the article's Creative Commons license and your intended use is not permitted by statutory regulation or exceeds the permitted use, you will need to obtain permission directly from the copyright holder.

Medizinische Fakultät | Postfach | 06097 Halle (Saale)

Medizinische Fakultät
der Martin-Luther-Universität
Halle-Wittenberg

To the Editor and Reviewers

**Institut für Medizinische
Immunologie**
kommissarische Direktorin:
Prof. Dr. Heike Kielstein

Revised Manuscript “The breast cancer tumor microenvironment in women living with HIV has an altered T cell infiltration profile independent of blood CD4⁺ T cell counts”

Hausanschrift:
Medizinische Fakultät
Magdeburger Str. 2
06112 Halle (Saale)

[www.umh.de/einrichtungen/
Institute/medizinische-
immunologie](http://www.umh.de/einrichtungen/Institute/medizinische-immunologie)

Dear Editor,
Dear Reviewers,

Sekretariat

N. Ott
Tel. +49 (0)345 / 557-1357
Fax +49 (0)345 / 557-4055
immunologie@uk-halle.de

please find enclosed our revised manuscript “The breast cancer tumor microenvironment in women living with HIV has an altered T cell infiltration profile independent of blood CD4 T cell counts”.

Forschungssekretariat

M. Heise
Tel. +49 (0)345 / 557-5041
Fax +49 (0)345 / 557-4055

Based on the comments of the reviewers our paper improved a lot. We did address all the queries raised by the reviewers, which will be summarized below:

Durchflusszytometrie

PD Dr. D. Riemann
Tel. +49 (0)345 / 557-4443
Fax +49 (0)345 / 557-1845

Aktenzeichen: -

Datum: 15.02.2025

Dear Reviewer #1,

Thank you very much for your interesting comments that improved the quality and the importance of this manuscript very much.

Please find the point-by-point answers below.

The study investigates the TME of BC in women living with HIV (WLWH) in Southern Africa. It compares immune cell composition and immune modulatory molecule expression between HIV-positive and HIV-negative BC patients. The findings suggest that HIV-positive patients exhibit distinct TME characteristics, including higher CD8⁺ T cell infiltration, increased CD276/B7-H3 expression, and more pronounced IFN- γ signaling, all of which are independent of age, stage, and hormone receptor status. These alterations contribute to poorer OS in HIV-positive BC patients. The study addresses a critical public health issue by focusing on the intersection of HIV and breast cancer, particularly in a region with high HIV prevalence. The study utilizes a robust dataset from the ABC-DO and SABCHO cohorts, which includes detailed clinical, epidemiological, and follow-up data.

The use of histomorphology, RNA expression analysis, and multiplex immunohistochemistry (MSI) provides a comprehensive assessment of the TME.

The study provides novel insights into the immune landscape of BC in HIV-positive patients, highlighting potential biomarkers and therapeutic targets such as CD276/B7-H3.

While the sample size (296 patients) is adequate, the findings may not be generalizable beyond the specific populations studied (South Africa and Namibia).

The study mentions survival analysis but does not provide detailed statistical models or hazard ratios for various subgroups.

The study controls for age, stage, and hormone receptor status but might benefit from a more detailed analysis of other potential confounders such as socio-economic status and treatment regimens.

While the study correlates immune profiles with survival, it lacks functional studies to elucidate the mechanistic pathways underlying the observed differences in the TME.

The study concludes that the altered T cell composition and CD276/B7-H3 expression in HIV-positive patients contribute to inferior survival and could be used for targeted treatment. However, it does not consider other potential factors for reduced OS such as the limited life expectancy due to HIV itself or suboptimal oncological treatment given the concomitant HIV. These factors should be mentioned and discussed by the authors.

Recommendations:

1. Include more detailed survival analysis with multivariate models to strengthen the association between immune profiles and outcomes.

Thank you very much for this recommendation. We included both treatment regimens (radiation, chemotherapy and antihormonal treatment) as well as socio-economic data (wealth score, number of children, employment status) if available and applied those factors in a multivariate analysis (survival analysis and regression analysis) and edited the figures accordingly.

Figure 1E – univariate Cox regression

E

Figure 1F – multivariate Cox regression

H

2. Discuss the findings in the context of other regions with high HIV prevalence to enhance the generalizability of the results.

We addressed this recommendation in the discussion as follows: Since the global prevalence of WLWH receiving HAART is increasing over time, non-AIDS-related cancers, such as breast cancer (BC), have become a significant cause of death among these patients (PMID: 38280393). Therefore, the analysis of the TME in BC lesions of WLWH and its influence on patients' outcome is not only interesting in the context of SSA, but also worldwide.

3. Consider including or suggesting future studies that explore the functional mechanisms behind the altered TME in HIV-positive BC patients.

Thus, an increased understanding of the functional mechanisms of T cell exhaustion in the context of cancer in general, but in particular also in cancer patients living with HIV, is urgently needed and should be addressed in future studies. This should include a more comprehensive analysis of both the TME and systemic immune status to enhance our understanding of the influence of chronic HIV in non-AIDS-related cancers.

4. Discuss other potential factors contributing to reduced OS in HIV-positive BC patients, such as life expectancy limitations due to HIV and potential suboptimal cancer treatments due to concurrent HIV infection.

Furthermore, premature aging caused by the HIV infection and drug toxicity by HAART and anti-cancer therapy including HAART-related liver damage are contributing factors that should be considered in future studies.

The manuscript presents valuable findings on the interplay between HIV infection and breast cancer, offering insights that could pave the way for targeted therapies and improved management of BC in HIV-positive patients, but it requires some additional work and clarifications to meet the high standards of Nature Communications.

These improvements will strengthen the manuscript and enhance its overall contribution to the field.

Dear Reviewer #2,

Thank you very much for your valuable comments that improved this manuscript very much.

Please find the point-by-point answers below.

Marcus Bauer et al. aimed to assess whether the HIV status influences the tumor microenvironment (TME) in breast cancer and mediates their poorer prognosis. They found HIV positive BC patients had a shorter OS, which was associated with increased numbers of CD8+ T cells, increased protein expression of CD276/B7-H3 and a more pronounced IFN- γ signaling. They provide valuable information about immune ecology of BC in HIV-positive patients. But there are also some major concern about this research.

1. In univariate COX regression analysis, "HIV positive" does not significantly predict overall survival (OS), yet "HIV positive patients with decreased numbers of CD4+ T cells" do show significant predictive power (Fig. 1C). So why not utilize the latter in multivariate COX regression analysis for prediction? (Fig. 1F) It is recommended that all subsequent analyses should be stratified based on this parameter.

Thank you very much for your comment on the importance of the HIV status and patients' immune status. We stratified WLWH into two groups depending on blood CD4+ T cell numbers and viral loads. WLWH with suppressed HIV and high CD4+ T cell numbers ($>500/\mu\text{l}$) were classified as WLWH - virally suppressed (WLWH - VS) and all other WLWH were called WLWH immune suppressed (WLWH - IS). We re-evaluated all downstream analyses and showed e.g., RNA expression, T cell subsets in the TME for HIV negative BC patients versus WLWH - VS and WLWH - IS.

2. Based upon the previous question, the authors state: "HIV-positive BC patients had a shorter OS (HR=1.25), which was associated with increased numbers of CD8+ T cells compared to HIV-negative BC patients." However, the abundant infiltration of CD8+ T cells was always reported to be positively associated with better survival. How to explain this controversial issue?

This controversial observation is indeed one of the most interesting points. While the number of CD8+ T cells was associated with better patients' outcomes in various tumor types and populations, we found no survival benefit in BC patients with higher CD8+ T cell numbers. This is due to the fact, that higher numbers of CD8+ T cells were found in WLWH, which have been shown to have a shorter OS compared to HIV negative BC patients. Moreover, based on the above-mentioned stratification of WLWH, we found the highest values of CD8+ T cells in WLWH - IS. We also found a higher expression of different immune checkpoint molecules, e.g., CD276, CTLA4 and LAG3 in WLWH. Those markers have previously been shown to be associated with T cell exhaustion. Higher numbers of exhausted T cells have also been linked with worse patients' outcome before. We included this also in the discussion (also shown in the answer of the next question)

3. Moreover, there is no correlation between intra-tumoral CD8+ T cells and peripheral blood CD4+ T cell counts. Yet, the authors suggest that peripheral blood CD4+ T cell counts and increased numbers of CD8+ T cells are both stronger predictors of shorter OS. Is there any contradiction here? Or are there other critical factors that have not been taken into account?

CD4+ T cell numbers in the blood are predictors for shorter OS due to patients' general immune status. In contrast, CD8+ T cell do not have a prognostic value in our cohort. This is in contrast to many studies that showed a favorable prognosis in BC patients with high CD8+ T cells in the BC TME, while a poor outcome was shown in patients with increased numbers of exhausted T cells. We included this issue also in the discussion:

Of note, higher CD8 expression in BC samples from WLWH has been previously associated with an improved outcome of BC patients (37), but high numbers of exhausted T cells predicted shorter survival in some BC subtypes (38).

4. In Fig. 3H, high expression of CD4 indicates a favorable prognosis in breast cancer, yet neither HIV status nor CD8 expression emerges as an independent predictive factor, contradicting the authors' conclusions.

The favorable prognostic value of higher CD4 expression is due to its association with the HIV status. As previously suggested, we edited the manuscript and grouped WLWH into patients with suppressed immunity and with viral suppression, respectively. However, our analysis shows that well established favorable prognostic factors in BC (TILs and CD8+ T cell numbers) lose their predictive value in this patient cohort. In contrast, patients with higher CD4 expression in the TME have a better prognosis, while a higher expression of CD276 is an independent predictor for worse outcomes.

5. The tumor sections require a re-evaluation for quantification. In Fig. 4I, there is no correlation between the proportion of T cell subsets and the percentage of TILs. In some cases, the total proportion of T cell subsets in a patient reaches up to 40%, while the TILs are less than 10%.

Thank you for the hint. We edited the image and double-checked the data shown in this Figure. It is noteworthy, since we used the whole slide images for the determination of TILs and selected areas for the MSI analysis (three areas in each slide, 3744 × 2808 pixel, 0.5 μm/pixel), some differences exist. Moreover, TILs encompass all immune cell subsets with rounded nuclei, while in the MSI analysis only T cell subsets were analyzed.

6. It is intriguing to note that HIV-positive patients exhibit a higher infiltration of CD8+ T cells within breast cancer. This phenomenon warrants further investigation. Are these CD8+ T cells antigen-specific, and do they target tumor antigens or HIV antigens?

For the current study, we only have formalin-fixed and paraffin-embedded tissues available and cannot make any statements about antigen specificity. In future studies, we would like to investigate T cell diversity and their immunological status using high-plex methods and sequencing.

7. HIV positive BC patients had a more pronounced IFN-γ signaling. Does the IFN-γ signaling pathway correlate with the prognosis of HIV-positive breast cancer patients?

To further investigate this observation, we used a published RNA data set and conducted GSEA. This showed that interferon gamma signaling, along with other immunological pathways, is increased in WLWH tumors.

Univariate Cox regression analysis revealed no prognostic value in the 48 samples analyzed.

Variable	reference	HR	CI95	p-value
pSTAT1	low pSTAT1	1.184	(0.44-3.19)	0.738
ISG15	low ISG15	2.184	(0.778-6.144)	0.139

8. Single cell sequencing and spatial profiling will provide a better understanding in the function of the cellular and non-cellular components of the TME contributing to tumorigenesis of BC in WLWH.

We can fully agree with this comment. In the future, we will carry out high-plex proteomic and spatial transcriptomic analyses.

9. The author only analyzed the sequencing data. It's suggested to investigate the function of TME cells, especially tumor-infiltrating CD4+ and CD8+ T cells, as well as the role of CD276, in breast cancer patients with HIV. The mechanisms underlying the increased infiltration of CD8+ T cells needs to be further explored.

High-plex proteomic analyzes and spatial transcriptomics analyzes would help us to answer these questions, but these technologies have not been available for us.

Dear Reviewer #3,

Thank you very much for your critical comments that improved this manuscript very much.

Please find the point-by-point answers below.

The study by Bauer and colleagues investigated the impact of HIV infection on the tumor microenvironment (TME) of breast cancer (BC) patients in Southern Africa. The main findings suggest that women living with HIV BC patients exhibit altered immune characteristics within the TME, including higher CD8+ cell counts, increased IFN-gamma and increased CD276 expression, which may contribute to their poorer prognosis compared to women without HIV.

The study does not present original questions or findings that advance the field. Critical information regarding patient immune status is not clearly provided, as patients living with HIV and those who are indeed immunodeficient are analyzed together as a single group, potentially leading to imprecise conclusions. Additionally, changes in CD4 and CD8 expression levels in HIV patients are already well-established, and data should include blood CD4 and CD8 levels. Thus, the differences in TME observed in the study may arise due to the fact that some HIV patients are immunodeficient, exhibiting low CD4 levels and high CD8 levels. Furthermore, the connection between immune exhaustion in cancer and HIV has been previously documented, perhaps except for data on CD276 expression. Therefore, while the study reinforces known associations, it does not significantly enhance our understanding of the interplay between HIV and breast cancer.

Thank you very much for this summary and the honest assessment of our results. Due to the importance of patients' systemic immune status, as pointed out by you as well as reviewer 2, we stratified WLWH into two groups depending on blood CD4+ T cell numbers and viral loads. WLWH with suppressed HIV and high CD4+ T cell numbers (>500/ μ l) were classified as WLWH - virally suppressed (WLWH – VS) and all other WLWH were called WLWH immune suppressed (WLWH – IS). We re-evaluated all downstream analyses and showed e.g., RNA expression, T cell subpopulations in the TME for HIV negative BC patients versus WLWH – VS and WLWH – IS.

To the best of our knowledge, we are the first to demonstrate an altered tumor microenvironment (TME) composition in a non-HIV-associated malignancy, with increased expression of immune checkpoint (ICP) molecules. Although an altered T cell composition could be anticipated based on the well-documented findings in the peripheral blood of individuals living with HIV undergoing HAART, it was unclear whether these changes would also be present in the TME and if they were linked to patient outcomes. Our findings indicate that the TME composition differs between women living with HIV (WLWH) and those without HIV, and we could demonstrate that some of the differentially expressed factors in the TME, such as CD4 and CD276, are also independent risk factors.

In the general comparisons between women living with HIV versus those who are not, the best method to see potential differences was to do a case-control study, matching as much as

possible (and for most variables) the two groups on a 1:3 or 1:2 basis. I don't see why the authors chose not to use this kind of study design.

Thank you very much for this comment. We agree that a case-control study design would be more favorable. We initially started our analysis with the cohort from South Africa, which closely approximates a 1:1 study. To determine if these findings were consistent in a different population, we then examined a subset of patients from the ABC-DO study from Namibia. Interestingly, we found similar results despite the significantly different proportions of women living with HIV (WLWH) versus HIV-negative patients.

*In general, the manuscript has important limitations. Several "significant" associations stated by the authors did not really reached statistical significance, but that didn't discourage the authors to state them in the manuscript. Just as examples, please see hazard ratios described in Figure 1F and stated in lines 292-5 of the manuscript, PD-1 expressin in Fig 5B and pStat1 expression in Fig 5C-E (listed in lines 357-61 of the ms). Differences in age of WLWH and those without infection listed in Table 1 and Fig. 1A are also non-significant, yet stated as such in the text. I also recommend that authors state exact p-values in their Figures, rater than "**", so we can assess the degree of significance in the associations pointed out. This is the case of Figures 3A, B, E, G, 4C, F, H, %B, D, as well Supplementary Fig. S1.*

All figures were edited and the p-values were given as numbers instead of asterisk. If p-values were >0.05 we did not state differences and instead used the term "trend of higher or lower expression".

In terms of format, the manuscript text requires extensive revision, since several erros and typos are present across the manuscrip. Also of particular importance, the authors should refraining from using "HIV-infected women" or "HIV-positive women" terms, which are not currently acceptable, and rather use "women living with HIV".

Thank you for this recommendation. We edited the manuscript and used the terms WLWH - virally suppressed (WLWH – VS) and WLWH - immune suppressed (WLWH – IS).

**Dear Reviewer #4,
We are glad that you assed this manuscript.**

Universitätsklinikum Halle (Saale) | Postfach | 06097 Halle (Saale)

Ching-yu Huang, PhD
Chief Editor for Immunology
Nature Communications

Universitätsmedizin Halle (Saale)

Institut für Pathologie
Direktorin:
Prof. Dr.
Claudia Wickenhauser

Hausanschrift:
Magdeburger Straße 14
06112 Halle (Saale)

Ihre Zeichen

Ihr Schreiben vom

Unsere Zeichen

Datum

07.04.2025

Sekretariat

Telefon: 0345 557-1281

Telefax: 0345 557-1295

pathologie@uk-halle.de

www.umh.de/pathologie

Revised Manuscript: “HIV status alters immune cell infiltration and activation profile in women with breast cancer”.

Dear Editor,

please find enclosed our revised manuscript “HIV status alters immune cell infiltration and activation profile in women with breast cancer”.

We included all files requested by you. We thank you very much for the opportunity to publish this study in Nature Communications.

Dear Reviewer #1,

Thank you very much for your feedback.

Dear Reviewer #2,

Thank you very much for your response.

Dear Reviewer #3,

Thank you very much for your critical comments that will help us for ongoing and future studies of the TME of WLWH.

Yours sincerely,

Marcus Bauer, Barbara Seliger and Eva Kantelhardt

Medizinische Fakultät
der Martin-Luther-Universität
Halle-Wittenberg